# Sparse Diffusion Policy: A Sparse, Reusable, and Flexible Policy for Robot Learning

**Yixiao Wang**[1*†], **Yifei Zhang**[1*], **Mingxiao Huo**[2*†], **Ran Tian**[1], **Xiang Zhang**[1], **Yichen Xie**[1],
**Chenfeng Xu**[1], **Pengliang Ji**[2], **Wei Zhan**[1], **Mingyu Ding**[1‡], **Masayoshi Tomizuka**[1]
[1]UC Berkeley    [2]Carnegie Mellon University

**Abstract:** The increasing complexity of tasks in robotics demands efficient strategies for multitask and continual learning. Traditional models typically rely on a universal policy for all tasks, facing challenges such as high computational costs and catastrophic forgetting when learning new tasks. To address these issues, we introduce a *sparse*, *reusable*, and *flexible* policy, Sparse Diffusion Policy (SDP). By adopting Mixture of Experts (MoE) within a transformer-based diffusion policy, SDP selectively activates experts and skills, enabling efficient and task-specific learning without retraining the entire model. SDP not only reduces the burden of active parameters but also facilitates the seamless integration and reuse of experts across various tasks. Extensive experiments on diverse tasks in both simulations and real world show that SDP 1) excels in multitask scenarios with negligible increases in active parameters, 2) prevents forgetting in continual learning of new tasks, and 3) enables efficient task transfer, offering a promising solution for advanced robotic applications. Demos and codes can be found in https://forrest-110.github.io/sparse_diffusion_policy/.

**Keywords:** Robot learning, Multitask and continual learning, Mixture of experts

## 1 Introduction

Generalist robots are gaining substantial attention in both academia and industry, capable of performing a wide range of tasks and continually learning new ones without losing previously acquired skills [1, 2, 3, 4, 5, 6]. Traditional approaches often rely on a universal and monolithic policy [1, 2] for all tasks, activating all the parameters in the large network for even simple tasks like pushing. Besides, given the diverse nature and lifelong requirements of robot learning tasks [7, 8], when encountering a new task, these approaches typically require costly fine-tuning [9] , which carries the risk of catastrophic forgetting of previously acquired skills. Task-specific adapters, such as LoRA [10], entail expanding active parameters during inference. An alternative approach is to train separate policies for different tasks, though this requires independent and from-scratch training for each task and prevents knowledge transfer across tasks.

Recent works on skill discovery [5, 11, 12] and chain of skills [13, 14] show promise in addressing the above challenges. These methods necessitate meticulous design with knowledge guidance such as visual features [5, 15, 16, 17, 18] and language prompts [14, 19], to learn different skills for different tasks, with the goal of reusing these skills in unseen scenarios. However, their skill abstraction modules are typically not scalable and the network structure is not designed to be sparse for efficient computing. As a result, the influence of network structure has not yet been thoroughly explored. Recently, Mixture of Experts (MoE) [20] has proven successful in large-scale applications across NLP, vision, and multimodal domains [21, 22, 23]. It selectively activates only a subset of expert

---

[*]Equal Contribution. [†]Project Lead. [‡]Corresponding Author.

8th Conference on Robot Learning (CoRL 2024), Munich, Germany.

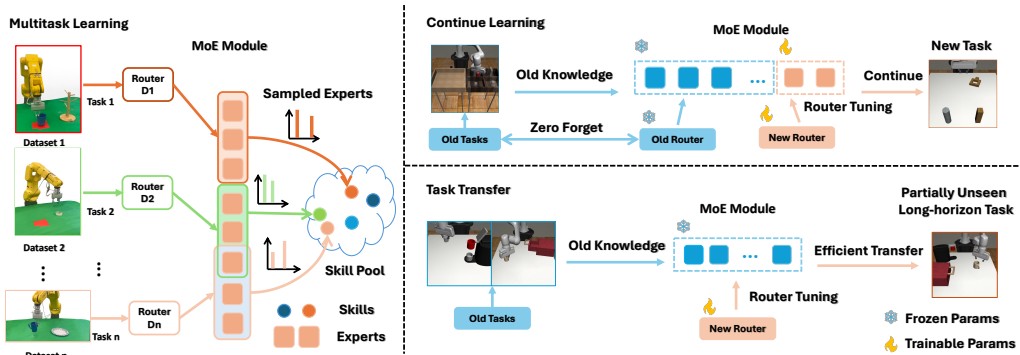

Figure 1: **Overview of Sparse Diffusion Policy (SDP).** 1) Multitask Learning: SDP can simultaneously acquire experts from different human demonstration datasets. Due to its sparsity, SDP can activate different experts for different tasks. Additionally, with its reusability, SDP can activate the same expert to share knowledge among tasks. 2) Continual Learning: With its flexibility, SDP can transfer to new tasks by adding only a few new experts to learn the new tasks. This approach mitigates catastrophic forgetting by retaining the old experts and routers. 3) Task Transfer: Leveraging its reusability, SDP can transfer to new tasks by tuning the old experts and routers for expert selection. This allows SDP to acquire new skills based on the previously learned knowledge.

networks selected by a router, allows experts to be utilized across various tasks and over time, and facilitates the integration of additional networks while preserving the functionality of existing ones. This observation raises a natural question: *Can the mere employment of a sparse, reusable, and flexible MoE structure overcome the challenge without the extensive integration of human-derived knowledge?*

Motivated by the above observation, we introduce Sparse Diffusion Policy (SDP), as depicted in Figure 1, a framework for multitask and continual learning by exploring the potential of integrating MoE architecture within a transformer-based diffusion policy [24]. SDP offers several advantages: 1) *Sparsity.* Only a select set of skills is activated at one time, significantly enhancing computational efficiency during inference. 2) *Reusability.* Skills are systematically reused across different tasks, for example, "pick and place" is a common skill frequently utilized in robotic tasks. 3) *Flexibility.* Skills for new tasks can be merged or added to the existing skill pool, enabling their flexible use in future tasks. We conceptualize the experts in MoE as specialized skills and the router as a skill planner (as illustrated in Figure 1). Furthermore, we explore specific training and application strategies of MoE for robotic learning.

Our extensive experiments in both simulations and real-world settings demonstrate the effectiveness of SDP in multitask, continual, and transfer learning for robotic tasks. It achieves superior multitask performance, with only a **1%** increase in active parameters compared to the single-task model. For continual learning, SDP maintains a higher success rate on new tasks without forgetting previously learned tasks, whereas the baseline model [10] requires activating over **62%** of its parameters. Furthermore, we investigate the potential for task transfer to complex, long-horizon tasks using a very small pretrained model, initially trained on only two half-length tasks. By training a highly lightweight router (less than **0.4%** of the total parameters), SDP outperforms models trained from scratch. Through experiments, we observe that the SDP is capable of extracting a broad range of skills through the combination of experts, and the router functions effectively as a skill planner.

## 2 Related Work

### 2.1 Multitask and Continual Learning in Robotics

In the field of robot learning, significant advancements have been made in multitask [25, 26, 27, 28, 29, 30, 13, 31, 32, 2, 33, 34, 35, 36, 37] and continual learning [38, 39, 40, 41, 5, 42], allowing robots to efficiently acquire and retain multiple skills over time. Multitask learning approaches,

such as policy distillation [43, 44, 45] and hierarchical reinforcement learning [46, 47, 48, 49], enable robots to learn and perform multiple tasks simultaneously by leveraging shared representations and decomposing complex tasks into manageable subtasks. However, these methods cannot induce sparsity during policy learning, which can enhance the efficiency of the policy network in multitask learning. Continual learning techniques, including regularization-based methods like Elastic Weight Consolidation (EWC) [50], memory-based strategies like experience replay, and architectural innovations such as Progressive Neural Networks (PNNs) [51], are developed to mitigate catastrophic forgetting, allowing robots to incrementally acquire new skills while retaining previously learned ones. Also, there are works in meta-learning [52, 53, 54] and few-shot learning [55, 56, 57, 58], that provide robots with the ability to quickly adapt to new tasks with minimal data. However, the MoE structure can naturally support continual learning without forgetting old tasks due to its unique architecture. This approach requires fewer additional techniques and can seamlessly integrate with multitask learning to create a dynamic task pool.

## 2.2 MoE in Computer Vision and Large Language Model

The Mixture of Experts (MoE) approach has seen significant advancements in both computer vision and large language models, offering a promising strategy to enhance model performance by leveraging specialized sub-models as "experts". In computer vision, MoE frameworks have been employed for multitask learning and transfer learning [59, 22, 60, 61, 62], demonstrating their efficiency in handling diverse and complex datasets such as segmentation, image classification. Moreover, many works [63, 64, 65] integrated MoE into the Transformer architecture, showing substantial gains in natural language processing tasks. These advancements underscore the potential of MoE systems to address the growing demands for computational efficiency [66, 67] and model accuracy [68, 69, 70] in both computer vision and language processing domains. This work focuses on leveraging the sparsity of MoE to conduct multitask and continue learning for the robot learning area. We also make full use of the MoE module to explore the efficient finetuning for task transfer.

## 3 Method

Our approach integrates Mixture of Experts (MoE) layers into a transformer-based diffusion policy network [24], combined with specifically designed training and application strategies for multitask and continual learning in robotics. Owing to the network's structural sparsity, we refer to our method as Sparse Diffusion Policy (SDP). In the subsequent sections, we first outline the problem formulation for multitask and continual imitation learning. We then discuss the integration of the Mixture of Experts (MoE) structure and explore how its *sparsity*, *flexibility*, and *reusability* can be specifically utilized for robot learning. Finally, we present the training strategies we have developed to further unleash the potential in the domain of robot learning.

### 3.1 Problem Formulation

We consider a set of robot tasks $\mathcal{C} = \{\mathcal{T}_j\}_{j=1}^J$. For task $j$, there are $N$ expert demonstrations $\{\tau_{j,i}\}_{i=1}^N$. Each demonstration $\tau_{j,i}$ is a sequence of state-action pairs. We formulate robot imitation learning as an action sequence prediction problem [24, 36], training a model to minimize the error in future actions conditioned on historical states. Specifically, for task $j$, imitation learning minimize the behavior clone loss $\mathcal{L}_{bc}^j$ formulated as

$$\mathcal{L}_{bc}^j = \mathbb{E}_{s_{t-o:t+h}, a_{t-o:t+h} \sim \mathcal{T}_j} \left[ \sum_{t=0}^{T} \mathcal{L} \left( \pi(a_{t:t+h}|s_{t-o+1:t}, \mathcal{T}_j; \boldsymbol{\theta}), a_{t:t+h} \right) \right]. \tag{1}$$

where $a$ is action, $s$ is state, $h$ is the prediction horizon, $o$ is the number of historical steps, $\mathcal{L}$ is a supervised action prediction loss such as mean squared error or negative log-likelihood, $T$ is the length of demonstrations and $\theta$ represents the learnable parameters of the network. In a multitask setting (to learn $\{\mathcal{T}_j\}_{j=1}^{J-1}$), the behavior cloning loss is given by $\mathcal{L}_{bc} = \sum_{j=1}^{J-1} \mathcal{L}_{bc}^j$. In

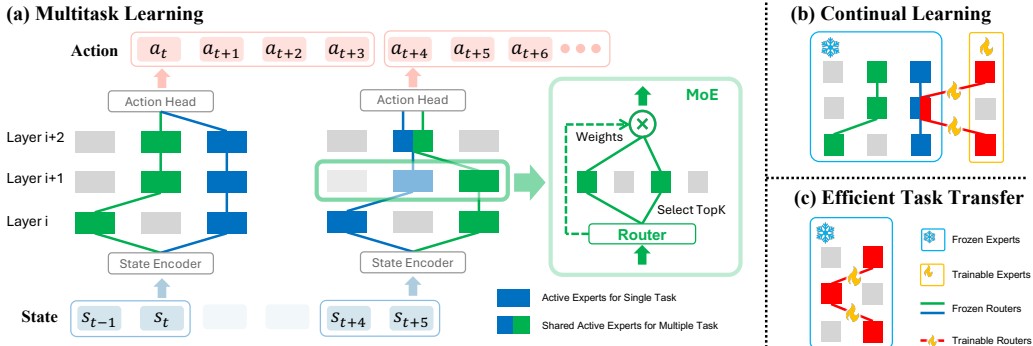

Figure 2: Network structure of SDP. SDP processes historical states, selectively activates specific experts via task-specific routers within each transformer layer, and ultimately outputs a sequence of future actions. In continual learning, SDP incorporates only a small number of trainable new experts along with a task-specific router. Leveraging the capabilities of previously acquired, expressive experts, SDP can quickly transfer to novel tasks by fine-tuning a lightweight router.

the case of continual learning, when only task $J$ is to be learned, we have access exclusively to the corresponding demonstrations $\mathcal{T}_J$ in this learning cycle, and the behavior cloning loss is $\mathcal{L}_{bc} = \mathcal{L}_{bc}^J$.

## 3.2 Sparse Diffusion Policy with MoE layers

We utilize the transformer-based diffusion policy [24] and replace the Feed-Forward Networks (FFN) with MoE layers. For the $n$-th MoE layer [20], $n = \{1, 2, ..., N\}$, there are $L$ experts $\{\mathcal{E}_l^n\}_{l=1}^L$ and one router $\mathcal{R}^n$. Each expert network $\mathcal{E}_l^n$ is composed of multilayer perceptrons (MLP). Router $\mathcal{R}^n$ compares the input $x \in \mathbb{R}^{1 \times M}$ with the expert embeddings $W^n \in \mathbb{R}^{M \times L}$ and get their scores. Only the Top-K expert networks are activated for inference. Specifically, the MoE layer output $y$ is derived as

$$y = \sum_{l=1}^{L} \mathcal{R}^n(x,l)\mathcal{E}_l^n(x), \quad \mathcal{R}^n(x,l) = \text{Top-K}(\text{Softmax}(xW^n), l). \tag{2}$$

where $\text{Top-K}(v, l)$ is the $l$-th element of vector $v$ if it is largest $K$ elements otherwise $0$.

**Multitask Learning.** Since different tasks require distinct experts for completion, we assign a task-specific router $\mathcal{R}_j^n$ to enable different tasks to select different experts based on the same historical states. As depicted in Figure 2, the same experts can be reused at different times within the same task and across various tasks, facilitated by the task-specific router and time-varied state. On the other hand, state-specific and task-specific experts can also be utilized and learned. More importantly, the activation of a limited number of experts demonstrates computational efficiency.

**Continual Learning.** MoE layers facilitate straightforward model expansion and support continual learning [61]. Specifically, for each new task, we freeze the previously learned experts and routers, integrate new trainable experts into each MoE layer and train the corresponding task-specific routers (See Figure 2). Catastrophic forgetting is mitigated by fixing previously learned parameters, thus enabling lifelong learning. Upon mastering the new task, the experts related to it are abstracted and become reusable in subsequent learning processes. Moreover, the computational cost remains constant despite the continuous integration of new tasks.

**Intuitive Interpretation and Task Transfer.** Intuitively, each unique combination of the TopK experts within every layer of a Mixture of Experts (MoE) architecture (refer to Figure 2) represents a distinct "skill". The routing mechanism acts as a planner for skill chains, selecting specific experts to assemble a skill. In this structure, the number of potential combinations of experts across $N$ layers is given by $\left(\frac{L!}{(L-K)!K!}\right)^N$. This formula suggests the capacity to cover a broad spectrum of diverse skills using a finite set of experts and layers. It also benefits continual learning. For instance, with parameters $L = 4, N = 2, K = 1$, adding one expert per layer (two in total) generates 9 new combinations. More promisingly, when confronted with an unseen long-horizon task, the inherent

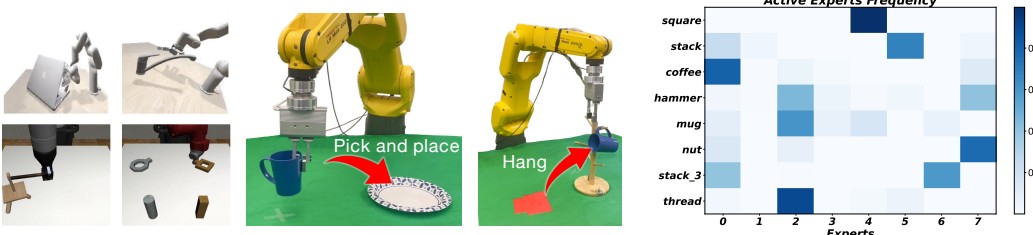

Figure 3: Task visualizations in 2D [74] & 3D [75] simulation and real robot experiments.

Figure 4: Expert selection frequency for each task in 2D simulation.

generalizability of MoE allows for significant flexibility. Even with only a few tasks previously trained, broad coverage of diverse skills by combination of experts makes it possible to train a new, lightweight router (merely 0.5% of the parameters discussed in Section 4.3) and behave well in a novel and even long-horizon task (denoted as $\mathcal{T}_J$ for convenience, the same in continual learning), enabling fast task transfer (See Figure 2).

## 3.3 Training Objective

Directly minimizing the behavior clone loss $\mathcal{L}_{bc}$ often leads to favoring certain experts and reinforcing their selection through a cycle of increased training and preference [71, 20]. To avoid overload of the expert, Previous studies [20, 63, 72, 73] have incorporated an auxiliary load balancing loss to prevent the overloading of any specific expert. However, in robot learning, certain skills are consistently utilized across various tasks, suggesting that some experts should be frequently engaged. For instance, pick-and-place skills are commonly required in numerous robotic tasks, resulting in excessive activation of the combination of experts related to the pick-and-place skill. Consequently, expert overload is typical in multitask robot learning, whereas task overload is atypical due to the distinct nature of each task. Each task includes specific components that necessitate unique skills for successful completion. Inspired by this observation, we propose encouraging experts to specialize in specific tasks [22, 61] to prevent task overload. Specifically, we would like to maximize the mutual information between the task $\mathcal{T}$ and the expert $\mathcal{E}^n$ for each MoE layer:

$$I(\mathcal{T}, \mathcal{E}^n) = \sum_{j=1}^{J} \sum_{l=1}^{L} p(\mathcal{T}_j, \mathcal{E}_l^n) \log \frac{p(\mathcal{T}_j, \mathcal{E}_l^n)}{p(\mathcal{T}_j)p(\mathcal{E}_l^n)} \tag{3}$$

where we assume that each task is equally important, i.e., $p(\mathcal{T}_j) = \frac{1}{J}$. Implementation details for $I(\mathcal{T}, \mathcal{E}^n)$ are presented in Appendix A. Thus, the total training objective is

$$\mathcal{L} = \mathcal{L}_{bc} - \gamma \sum_{n=1}^{N} I(\mathcal{T}, \mathcal{E}^n) \tag{4}$$

where $\mathcal{L}_{bc} = \sum_{j=1}^{J-1} \mathcal{L}_{bc}^j$ in multitask learning, and $\mathcal{L}_{bc} = \mathcal{L}_{bc}^J$ in continual learning and task transfer.

## 4 Experimental Results

### 4.1 Multitask Learning

In this section, we discuss *sparsity* and *reusability* of our SDP, highlighting its ability to simultaneously acquire multiple experts efficiently while sharing experts across tasks. As shown in Figure 3, we conducted three kinds of experiments: 2D vision-based simulations, 3D point cloud-based simulations, and real-robot experiments. Implementation details are presented in Appendix B.1.

**2D Simulation Results.** We evaluate the performance of SDP on 8 tasks in Mimicgen [74]. Mimicgen includes 1K-10K human demonstrations per task with broad initial state distributions, effectively showing the generalization for multitask evaluation. To our knowledge, this is the first work to explore multitask training on the Mimicgen benchmark. We choose task-conditioned diffusion

Table 1: Multitask evaluation on MimicGen [74]. We report average success rate of the best three checkpoints, total parameters (TP), activate parameters (AP). Our results are highlighted in light-blue cells.

| Method | TP(M) | AP(M) | Square | Stack | Coffee | Hammer | Mug | Nut | Stack Three | Thread | Avg. |
|---|---|---|---|---|---|---|---|---|---|---|---|
| TH | 52.6 | 52.6 | **0.76** | 0.98 | 0.72 | 0.97 | 0.63 | **0.52** | 0.73 | 0.55 | 0.73 |
| TT w/ 3Layer | 144.7 | 52.6 | 0.73 | 0.95 | 0.76 | **0.99** | 0.65 | 0.49 | 0.68 | 0.59 | 0.73 |
| TCD [76, 19] | 52.7 | 52.7 | 0.63 | 0.95 | 0.77 | 0.92 | 0.53 | 0.44 | 0.62 | 0.56 | 0.68 |
| Octo [77] | 48.4 | 48.4 | 0.68 | 0.96 | 0.72 | 0.97 | 0.48 | 0.32 | 0.72 | 0.64 | 0.69 |
| SDP | 126.9 | 53.3 | 0.74 | **0.99** | 0.83 | 0.98 | **0.70** | 0.42 | **0.76** | 0.65 | **0.76** |
| Light SDP | 53.3 | **38.7** | 0.75 | 0.96 | 0.83 | 0.97 | 0.55 | 0.50 | 0.74 | **0.73** | 0.75 |

Table 2: Multitask evaluation on DexArt [78] and Adroit [79].

| Method | DexArt [78] | | | | Adroit [79] | | | |
|---|---|---|---|---|---|---|---|---|
| | Toilet | Faucet | Laptop | Avg. | Door | Hammer | Pen | Avg. |
| TT w/ 1Layer | 0.73 | 0.35 | **0.85** | 0.64 | 0.63 | 0.92 | 0.54 | 0.70 |
| TCD [76, 19] | 0.72 | 0.33 | 0.80 | 0.62 | 0.63 | 0.83 | 0.42 | 0.63 |
| SDP | **0.75** | **0.43** | 0.82 | **0.67** | **0.70** | **0.97** | **0.58** | **0.75** |

(TCD) [76, 19], fine-tuning the action head (TH), fine-tuning last three transformer layers (TT w/ 3Layer), Octo [77] as our baselines. TCD and Octo require the activation of all network parameters, exhibiting a dense structure rather than a sparse one (Ours). We train our standard SDP model as well as a smaller version, referred to as Light SDP.

Each model is trained for 300 epochs using one A6000 GPU for 130-150 hours and evaluated every 50 epochs. We report the average success rate of the best three checkpoints in Table 1. Thanks to the *sparsity*, our SDP outperforms all baselines with the same level of active parameters in the policy network. Additionally, our Light SDP also surpasses all baselines with significantly fewer active parameters, while maintaining the same level of total parameters. Another observation is that sparse models (SDP, TH, TT) outperform dense models (TCD and Octo). This suggests that dense structures may hinder the learning of distinct policies across tasks, whereas sparse structures, which allocate separate parameters for different tasks, facilitate the learning of more diverse actions. Task-expert frequency map in Figure 4 shows that each task activates only a subset of the experts, contributing to computational efficiency. More importantly, the map reveals that different tasks can activate the same expert, demonstrating the *reusability* of experts across various tasks.

**3D Simulation Results.** We evaluate our approach on 3 tasks in DexArt [78] and 3 tasks in Adroit [79]. Specifically, we follow [75] and integrate the Mixture of Experts into the Feed-Forward Network blocks of the 3D diffusion policy network [75]. The number of active parameters is set to be equivalent to the original. We train for 6000 epochs and evaluate every 200 epochs. We report the average success rate of the best three checkpoints. As shown in Table 2, our method outperforms the baseline, showing the effectiveness of our SDP in 3D perception settings.

**Real Robot Experiment Results.** We conduct real robot experiments on FANUC LRMate 200iD/7L robotic arm outfitted with an SMC gripper. We choose three diverse and universal: pulling a circle, picking and placing a cup, and hanging a cup. For the first two tasks, we collect 20 demonstrations each, and for the last task, we collect 40 demonstrations with two distinct hanging sticks. More details can be found in the Appendix B.2. The robot is manipulated using admittance control [80], which can achieve compliant robot motion to ensure safety during manipulation. The number of training epoch is 2000. As shown in Figure 5, our method greatly outperforms TCD [76, 19]. Additionally, we found that TCD cannot clearly

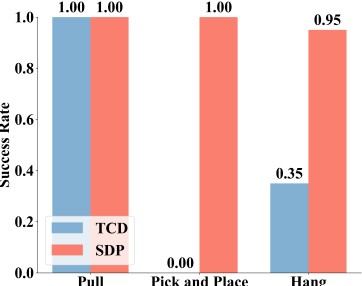

Figure 5: Multitask evaluation on real-world tasks.

distinguish the multimodal action distributions across different tasks. We hypothesize that this may be caused by insufficient sparsity. More details and visualizations can be found in the Appendix B.2.

Table 3: Evaluation on continual learning. Comparison of different policy decoders. *AP* denotes *Active Parameters* of the policy network. Grey blocks indicate performance on new tasks; light-blue blocks indicate performance on previous tasks.

| Method | Stage 1 | | Stage 2 | | | Stage 3 | | | |
|---|---|---|---|---|---|---|---|---|---|
| | Can | AP | Can | Lift | AP | Can | Lift | Square | AP |
| FFT | 0.97 | 9.0 M | 0.00 | 1.00 | 9.0 M | 0.00 | 0.00 | 0.89 | 9.0 M |
| LoRA[10] | 0.94 | 9.0 M | 0.94 | 1.00 | 12.0 M | 0.94 | 1.00 | 0.73 | 14.9 M |
| SDP (Ours) | 0.96 | 9.2 M | 0.94 | 1.00 | 9.2 M | 0.94 | 1.00 | 0.75 | 9.2 M |

## 4.2  Continual Learning

In this section, we evaluate the performance of the Sparse Diffusion Policy (SDP) in continual learning, with a focus on the framework's *flexibility*. When integrating a new task, we add a small number of experts and a new router to the Mixture of Experts (MoE) structure, while freezing the existing experts, routers, and other non-MoE modules. We then train only the newly added router and experts, enhancing the efficiency of learning new tasks and preventing catastrophic forgetting [81] of previously learned tasks. Assume we continuously learn three tasks, $a \to b \to c$. We first train the model on task $a$ (Stage 1), then on task $b$ (Stage 2), and finally on task $c$ (Stage 3), evaluating the performance on both the current and previous tasks.

**Comparison with baselines.** We choose three tasks from robimimic [82] for evaluation: Can $\to$ Lift $\to$ Square. First, we compare our approach with LoRA [10] and a fully fine-tuned (FFT) version of our method. Each method is trained for 500 epochs and evaluated every 50 epochs. We report the average success rate of the best three checkpoints. The results, presented in Table 3, demonstrate that our method generally exhibited superior performance in both new tasks and previous tasks. Conversely, FFT exhibited significant forgetting of previously learned tasks when learning new ones, whereas LoRA struggled with a substantial increase in the number of active parameters. These findings underscore the *flexibility* of our SDP.

**Ablation study on continual learning.** In this section, we report ablation study on mutual information (MI) loss in Section 3.3. We selected three tasks from MimicGen [74]: Stack $\to$ Hammer Cleanup $\to$ Coffee, where task complexity progressively increases (e.g., the Coffee task requires not only picking and placing but closing the coffee cap.). Since our SDP avoids catastrophic forgetting [81], we report only the performance of the newly added tasks in Figure 6. The results emphasize the significance of MI loss. We argue that the router initially favors previously trained experts, as newly added experts, being untrained, produce random actions, reinforcing reliance on frozen experts. However, MI loss encourages the router to select task-specific experts,

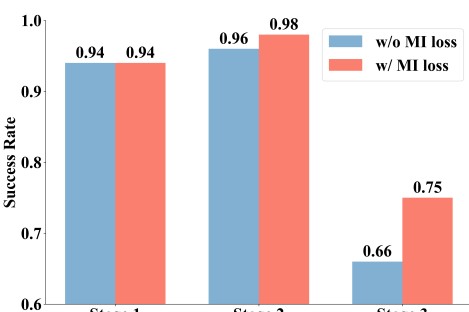

Figure 6: **Ablation Study on MI loss.** We report the success rate for the new tasks, as the performance on previous tasks remains the same thanks to our SDP structure.

promoting the use of seldom-selected experts. Additional ablation studies are presented in Appendix C.3, where experiments further demonstrate the critical role of comprehensive MoE structures and MI loss in achieving optimal performance in complex continual learning settings.

## 4.3  Efficient Task Transfer

In this section, we evaluate the *reusability* of our SDP, highlighting its ability to efficiently transfer to new tasks by learning how to leverage previously acquired experts (router tuning). Additionally, we visualize the evolution of expert scores to assess the role of prior skills in enabling the transfer to new tasks.

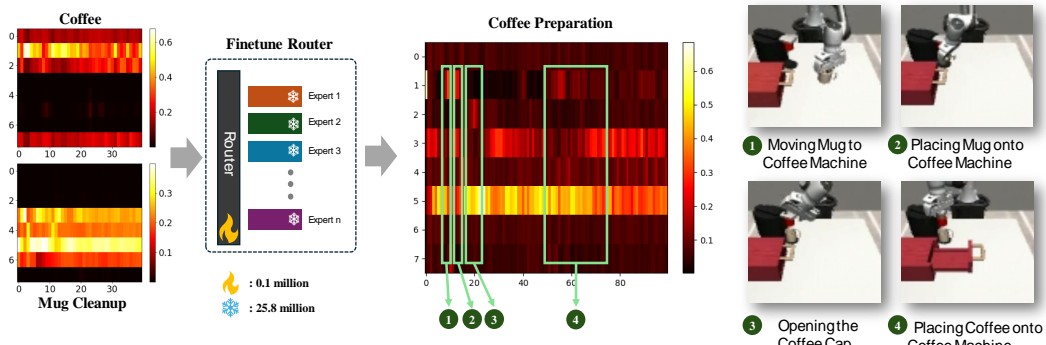

Figure 7: **Visualization of Expert Scores versus Timesteps.** When the robot interacts with the coffee machine (not encountered in Mug Cleanup), experts trained on the Coffee achieve higher scores and are activated, offering interpretable insights into the SDP. Also, our model only requires less than 0.1M (**0.5%**) trainable parameters for efficient task transfer.

**Efficient task transfer experiment.** We first train our SDP on base tasks, then freeze all experts and fine-tune only the router for the new task. Specifically, we use Coffee and Mug Cleanup as the base tasks and Coffee Preparation as the new task, all from MimicGen [74]. Coffee Preparation is a long-horizon, partially unseen task requiring unique skills, such as moving the mug to the drip tray. After pretraining on the base tasks, we fine-tune only the MoE router and the vision encoder for the new task. For comparison, we also train a model with the same structure from scratch. Both models are trained for 100 epochs. As shown in Table 4, our method uses less than **0.5%** of the parameters compared to the train-from-scratch approach while achieving better performance, demonstrating that the experts learned from the base tasks effectively cover diverse skills and offer strong representation for distinct tasks.

**Skills visualisation.** As shown in Figure 7, we visualize the evolution of expert scores on the Coffee, Mug Cleanup, and Coffee Preparation tasks. We observe that when knowledge from Coffee is required but not present in Mug Cleanup, the experts trained on Coffee are activated. Four typical examples are illustrated on the right of Figure 7. By composing

Table 4: Evaluation on Coffee Preparation [74]. We report the number of trainable parameters (Train. Params) in the policy network.

| Method | Train. Params | Success Rate |
|---|---|---|
| Scratch | 25.9M | 0.70 |
| Rou. only | **0.1M** | **0.80** |

these experts, SDP can acquire new skills, such as moving the mug to the coffee machine's drip tray. Additionally, we find that some experts are consistently activated in Coffee Preparation, while others are seldom used. This insight suggests the possibility of policy distillation, which we leave for future work. More experiment details and explanations are presented in Appendix D.

## 5 Discussion and Limitation

In this paper, we have introduced the Sparse Diffusion Policy (SDP) framework, which integrates Mixture of Experts (MoE) layers into the diffusion policy. Our approach is designed around three key principles: *sparsity*, *flexibility*, and *reusability*. By activating only relevant portions of the network for specific tasks, SDP can induce the sparsity for efficient multitask learning. The flexibility of our algorithms is capable of acquiring new tasks without forgetting existing skills, while the reusability of existing knowledge allows for efficient multitask and task transfer learning. Our model has great potential for future large-scale robot learning.

**Limitation.** Our SDP may fail if the shared knowledge in the network is too limited but same experts are activated by different routers. Additionally, the router in SDP is task-specific, which hinders the ability for universal task completion. By combining it with large language models, the SDP conditioned on language with a broader policy can be applied to a wider range of robot learning scenarios in the future.

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

# Appendix

## A  Mutual Information Loss

In order to calculate $I(\mathcal{T}, \mathcal{E}^n)$, we need to get $p(\mathcal{T}_j)$, $p(\mathcal{E}_l^n)$ and $p(\mathcal{T}_j, \mathcal{E}_l^n)$. $p(\mathcal{E}_l^n)$ can be calculated by counting the number of selection times of expert $l$ across all the tasks. We assume each task is equally important so $p(\mathcal{T}_j) = \frac{1}{J}$ where $J$ is the number of tasks. For $p(\mathcal{T}_j, \mathcal{E}_l^n)$, we have $p(\mathcal{T}_j, \mathcal{E}_l^n) = p(\mathcal{T}_j)p(\mathcal{E}_l^n|\mathcal{T}_j) = \frac{1}{J}p(\mathcal{E}_l^n|\mathcal{T}_j)$. $p(\mathcal{E}_l^n|\mathcal{T}_j)$ can be calculated by counting the number of selection times of expert $l$ for task $j$.

## B  Multitask Learning Experiments

### B.1  Implementation Details

For multitask learning in 2D simulation environments, we train the policy for 300 epochs using 12 transformer blocks with 512 embedding dimensions. The batch size is set to 64, and the learning rate is 0.0001, optimized with Adam [83]. During evaluation, we use 2 observation steps, followed by 8 action planning steps, executing only the first step. For SDP, we use 8 experts, activating 2 experts per task, with each expert having the same size as the original Feed-Forward Network (FFN). In Light SDP, we increase the number of experts to 16 and activate 8 experts, with the size of each expert reduced to $\frac{1}{16}$ of the original FFN to maintain the overall model size.

For the 3D simulation environments, we use 2 layers of transformer blocks with 256 embedding dimensions. The learning rate is 0.0001, and we use the Adam optimizer [83].

### B.2  Real Robot Experiments

In this section, we provide additional details on the real-world robot experiments, as depicted in Figure 9. For the pull task, we collected 20 human demonstrations, where the objective is to pull the circle to the red region at the center of the table. For the pick-and-place task, we also gathered 20 demonstrations, where the goal is to pick up the cup and place it on the plate. Lastly, for the hang task, we collected 20 demonstrations for each of the two goal points. The initial object position is used as the state input for the Pull and Pick-and-Place tasks, while the goal point serves as the state input for the Hang task. All state inputs are encoded using learnable Fourier embeddings.

We observe that TCD [76, 19] struggles to capture the multimodality of the task-specific policies and distinguish between different task behaviors, leading to failures. In detail, TCD always rotates the end effectors in the Pick and Place (See Figure 8), a behavior not observed in the demonstrations. Instead, rotations are necessary only for Hang. This observation indicates that TCD tends to conflate policies from different tasks and struggles to capture multimodal action distributions across diverse tasks.

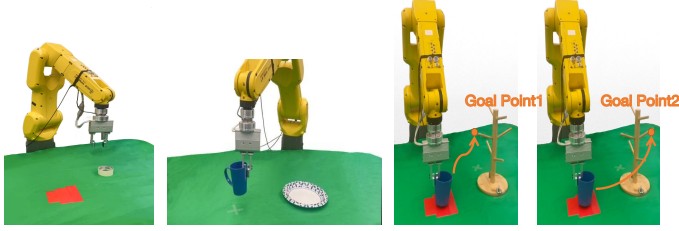

Figure 8: The visualizations for the real-world experiments are shown in the figures from left to right, representing: Pull, Pick-and-Place, Hang (goal point 1), and Hang (goal point 2).

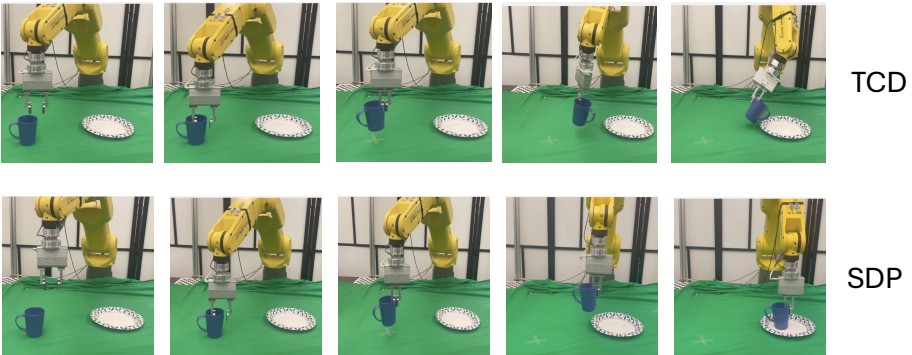

TCD

SDP

Figure 9: The baseline model fails because it cannot distinguish among different tasks.

# C  Continual Learning Experiments

## C.1  Implementation Details

For the continue learning setting, we use 8 layers of transformer blocks with 256 embedding dimensions. The learning rate is 0.0001, and we use the Adam optimizer [83]. For the SDP policy, we set the number of experts for a new trainable task to 8, and activate 2 experts for each new task.

## C.2  Comparison with Baselines

In this section, we conduct additional continual learning experiments using a fully fine-tuned visual encoder. As shown in Table 5, after transferring to the final task (Square), our model outperforms the LoRA method in terms of transferability with the fully fine-tuned visual encoder. However, fully fine-tuning the visual encoder results in complete forgetting of previous tasks. Therefore, utilizing a comprehensive set of MoEs is essential for effective continual learning.

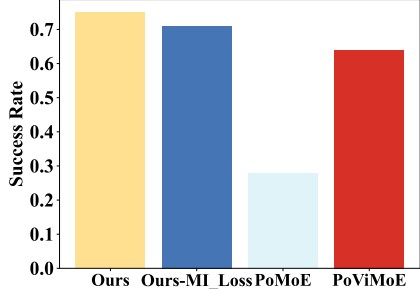

Figure 10: **Ablation Study for Continual Learning.** *Ours-MI_Loss* refers to our method excluding the Mutual Information loss. *PoMoE* indicates our method without any other Mixture of Experts (MoE) besides the Policy MoE. *PoViMoE* denotes our method with only the Vision MoE and Policy MoE, excluding all other MoEs.

## C.3  Ablation Study

In this section, we present ablation study results on three easy tasks: Can → Lift → Square. Can involves picking and placing a can, Lift focuses on lifting a cube, and Square requires inserting a square peg into the correct post. We report the performance on the Square task at Stage 3 in Figure 10. These results highlight the importance of employing a comprehensive set of MoEs and the Mutual Information loss to achieve optimal performance in complex continual learning scenarios. Compared to the results in Figure 6, we observe that in more challenging continual learning settings (Stack → Hammer Cleanup → Coffee), Mutual Information loss plays a more significant role in learning new tasks.

# D  Efficient Task Transfer

Recall that Coffee and Mug Cleanup are the base tasks, Coffee Preparation is the new task. In Coffee, the robot is required to place the pod into the holder and close it. For Mug Cleanup, the robot must open a drawer, place the mug inside, and close the drawer. Coffee Preparation involves

Table 5: Evaluation on continual learning. Comparison of different policy decoders. Grey blocks indicate performance on new tasks; light-blue blocks indicate performance on previous tasks.

| Vision Encoder | Policy Decoder | Stage 1 Can | Stage 2 Can | Stage 2 Lift | Stage 3 Can | Stage 3 Lift | Stage 3 Square |
|---|---|---|---|---|---|---|---|
| FFT | LoRA [10] | 0.97 | 0.00 | 1.00 | 0.00 | 0.00 | 0.57 |
| FFT | MOE (Ours) | 0.93 | 0.00 | 1.00 | 0.00 | 0.00 | 0.75 |

placing the mug into the drip tray, opening the holder, and the drawer, placing the pod into the holder, and then closing it. This task can be seen as a composite of Coffee and Mug Cleanup, but it includes unique actions, such as moving the mug to the drip tray, which are not present in the initial training tasks.

Our task is to leverage the frozen experts developed from the previous two tasks and learn task-specific routers that combine these experts to acquire new skills (e.g., moving the mug to the drip tray). To illustrate the router's functionality, we aim to activate different experts for each base task. Thus, we set eight experts per layer and activate only the top two. We visualize Expert Score versus Timesteps in the final transformer layer at the last fifth diffusion timestep in Figure 7. We observe that Coffee primarily activates experts 0127, while Mug Cleanup activates experts 3456, indicating no shared experts between these tasks. This allows us to identify which base task's knowledge is being used for the new task, Coffee Preparation.

We observe that experts trained on the Coffee task are more likely to be selected when the task requires information specific to the coffee machine, which does not appear in the Mug Cleanup task. For example, Coffee-related experts are activated when the action involves placing the pod. This observation highlights that the router functions as a skill planner, with experts serving as individual skills. The router effectively composes these skills across tasks to tackle complex and previously unseen tasks.

Interestingly, we find that new skills can emerge through the combination of experts from different tasks. For example, when the robot moves the mug to the coffee machine's drip tray—a new skill—it must determine the coffee machine's location, triggering the activation of Coffee-related experts. In most cases, however, experts from Mug Cleanup are activated, indicating that their combination suffices for the majority of Coffee Preparation, except for actions that specifically require Coffee-related information. This demonstrates the broad skill coverage achieved through expert composition, underscoring the expressive power of our SDP. Additionally, we observe that some experts are seldom activated. Given the frequency of activation across tasks, we can prune, merge and enlarge the experts, suggesting a potential connection between our SDP and policy distillation, which we leave for future work.

Based on the discussion above, aligning with the intuitive explanation in Section 3.2, the experts could be viewed as a pool of skills, with the router functioning as a chain planner, enabling our SDP to efficiently transfer knowledge to new tasks.

