# OpenReview forum: "Sparse Diffusion Policy:  A Sparse, Reusable, and Flexible Policy for Robot Learning"
_robot-learning.org/CoRL/2024/Conference — CoRL 2024_

### Official Review · Reviewer_ntiJ · 2024-07-12
**A very clear contribution overall, requires minor improvements**

**Originality:** 3
**Technical Quality:** 4
**Clarity Of Presentation:** 5
**Potential Impact:** 3
**Recommendation:** 4
**Confidence:** 3

**Review:**

The paper proposes a sparse (computationally efficient) diffusion policy, which is flexible in its knowledge addition and adaptation, and reusable by simply switching the skill selection model. The work would be stronger if it presented comparisons with more standard baselines.

Strengths:

- The authors have released the code and demonstrations used, significantly strengthening the reproducibility of the method’s results, as well as making the work more convincing.

- Strong results supporting all claims made by the paper.

- The writing is clear and concise.

Weaknesses.

- Brief related works section does not cover many relevant works.

- The future works section is lacking.

- Some method details are unclear, e.g. how are the expert embedding obtained. While the code is provided, based on the paper alone, there might not be enough information provided to reproduce the work.

- Justification of the diffusion-based model is insufficient.

- While the chosen baselines are good, it would be better to include further baselines from multi-task learning and continual learning.

**Quality Of The Limitations Section:**

3

**Questions For Rebuttal:**

Other comments:

Very good strong motivation and introduction.

Fig 2. It’s unclear what the gray blocks represent exactly.

The comparison with LoRA does not show significant difference between the proposed method and LoRA, while L218 claims that it “struggled”. The results presented do not seem to support that statement.

Fig. 6. It is unclear what the heatmap axes are.

While diffusion policies have gained traction in recent years, it is unclear why 3D diffusion policy was chosen as the particular – could any model be used as the backbone? Could there be a mix of differently structured experts? Is “diffusion” core to the contribution? If yes, it asks for more justification and related works. If not, it is unclear why the contribution is focused around it.

The reasoning behind some of the references is unclear – Table2 / Fig 3. refers to previous works, but do the depicted tasks originate from those works?

One strength of the method which is not disccussed is interpretability. The skill-based router model allows for more interpretability of the model’s behaviour – given specialized experts it would be clear what the model “intends”.

**Robotics Focus:**

4

**Summary Of Paper:**

The paper proposes a sparse (computationally efficient) diffusion policy, which is flexible in its knowledge addition and adaptation, and reusable by simply switching the skill selection model.

**Summary Of Recommendation:**

I recommend the work is accepted due to clear strengths. However, during the rebuttal phase I could be swayed based on author responses and reviewer comments.

---

### Official Review · Reviewer_1A92 · 2024-07-19
**Appealing idea but results not convincing**

**Originality:** 3
**Technical Quality:** 1
**Clarity Of Presentation:** 2
**Potential Impact:** 3
**Recommendation:** 3
**Confidence:** 4

**Review:**

I understand the appeal of the proposed technique and agree that using a mixture of experts can lead to the reuse of skills for multitask/continual learning. Although MoEs with transformer-based architectures have been used in other domains, it is not trivial to transfer this to robotics, and the proposed SDP framework can be potentially useful for the robot learning community. However, there are several open questions and I think the clarity of the paper needs to be improved, especially in the part where SDP is described. The biggest cause of concern for me is related to the significance of some of the results, where the difference between the proposed method and other baselines is not very clear (see rebuttal questions below).

*Strengths*:
1. Appealing idea. MoE is potentially a good way to acquire new manipulation skills without catastrophic forgetting and unbounded parameter growth.
2. Addresses both multitask and continual learning scenarios

*Weaknesses*:
1. The paper lacks clarity in several places
2. Results do not clearly show that the proposed method is better than other baselines
3. Many open questions

**Quality Of The Limitations Section:**

3

**Questions For Rebuttal:**

1. Line 76: Please clarify what is meant by "additional techniques"

2. Line 115: Please explain how the Router $\mathcal{R}^n$ compares the input $x$ with the expert embeddings $W^n$. How are the embeddings trained?

3. Fig. 2(a) is confusing. Some questions related to this:
(i) What do the colors (blue, green, red) denote?
(ii) Does each rectangle denote a single MLP expert?
(iii) I assume each row of boxes (green border) is a single MoE layer. If an MoE layer produces a single output, what is the significance of the outputs between blue and blue and between green and green boxes?

4. Please state in the caption what the flame symbol signifies in Figs 2(b) and (c)

5. Eq. (2) does not match Eqs. 2, 3 from [20]. Please explain the difference.

6. Line 127 states "[we] integrate new trainable experts into each MoE layer and train the corresponding task-specific routers". Do you have multiple routers in each MoE layer or only a single router per layer (as shown in Fig. 2)?

7. Line 131: In what sense does the computational cost remain constant (storage, FLOPS, or something else)? As you add some experts and a router for each task, does the cost remain **nearly** constant or absolutely constant?

8. In equation (3), how is $p(\mathcal{T}_j, \mathcal{E}^n_l)$ computed?

9. Please consider commenting on the suitability of the baselines used in the experiments (i.e. why did you choose the baselines used in the paper?).

10. In Table 1, the average of the best three checkpoints is presented. Is each checkpoint an independent trial? Why do you use the best three and not the average over all checkpoints?

11. In the results reported in Tables 1, 2, and 4, the difference between the proposed method and other baselines is quite small. These results are the average over the three best trials (as far as I understood) and the authors do not report the standard deviation. Given the small number of checkpoints, the difference between the methods may not be significant. It is difficult to be convinced that the proposed method is superior to the baselines without considering more checkpoints and looking at the average and SD, or conducting explicit statistical tests.

12. In Table 1, does the phrase "active parameters" refer to the parameters belonging to the experts selected by the router (plus, other shared parameters)? Please also report the total parameter size.

13. Line 180: A quantitative comparison of computational efficiency would help the reader. If I understand correctly, the active parameters of the sparse method (SDP) are nearly the same as the other non-sparse baselines. Why then is SDP more efficient computationally?

14. Please consider commenting on the reason for the very long training times needed by your models. Do all the baselines need to be trained for similar durations?

15. For the ablation study (line 223), why do you only report the results for the last task? Since this is a CL ablation, the results of all tasks should be reported. If space is a constraint, consider reporting the full results in the supplementary material/appendix.

Some minor comments:

1. Many citations point to the Arxiv versions of published papers. Please change the citations to the published versions. Some examples (this is not an exhaustive list):
- Ajay et al. Is conditional generative modeling all you need for decision-making - Published in ICLR 2023
- Liang et al., Skilldiffuser: Interpretable hierarchical planning via skill abstractions in diffusion-based task execution - Published in CVPR 2024
- Chi et al., Diffusion policy: Visuomotor policy learning via action diffusion - Published in RSS 2023

2. Tables 1,2,3 use "SDP (ours)" and Table 4 uses "MoE (ours)". Please be consistent.

3. In the caption of Table 1, "activate" should be "active"

4. Line 88: "continue" should be "continual"

**Robotics Focus:**

4

**Summary Of Paper:**

This paper presents a technique named Sparse Diffusion Policy (SDP) for multitask and continual learning of manipulation skills. SDP utilizes a transformer-based diffusion policy where the final layers of the model are composed of Mixture-of-experts layers, where each expert is an MLP. Task-specific trained routers activate different combinations of experts for different tasks. The reuse of experts across tasks leads to the reuse of learned skills without unbounded growth in parameters. Experimental results include multitask and continual learning scenarios in simulation and the real world.

**Summary Of Recommendation:**

The idea of using MoE layers to avoid catastrophic forgetting while continually learning manipulation skills is appealing. However, the results do not fully support the authors' claims about the superiority of their approach. Moreover, several questions remain unanswered by the paper in its current form.

---

### Official Review · Reviewer_bbJc · 2024-07-20
**A promising idea for multi-task and continual robot learning with difussion policy**

**Originality:** 4
**Technical Quality:** 3
**Clarity Of Presentation:** 3
**Potential Impact:** 3
**Recommendation:** 3
**Confidence:** 3

**Review:**

### Strengths
- The problem tackled by this work is of great importance in the imitation learning/robot learning community;
- The proposed idea of equipping the Mixture-of-Experts (MoE) layer to the policy network is highly sensible and feasible;
- The presentation of this work is clear and easy to follow;
- The experiments are comprehensive, covering all the claimed properties of the model.

### Weaknesses/ Suggestions
 - The first question is about the claimed **sparsity**. In experiments such as Tables 1 and 2, the proposed method has similar active parameters with the baselines, while active parameters denote only a subset of the experts in the proposed model and the whole one for the baseline models. In this sense, the parameters of the whole model (all experts) to be stored would be much more than the baselines. Does this cause the problem of high memory consumption? Should a proper comparison be made with the same amount of the **overall** model for both the proposed method and the baselines?
 -  The tasks evaluated in the experiments are insufficient in terms of amount and task diversity [1, 2]. There are only 7 tasks in 2D simulation, 3 tasks in 3D simulation, 3 tasks on real robots for multi-task learning, and 3 similar tasks for continual learning. With the marginal improvement in the success rate, the experimental results are less convincing.
 - To better support the claim in the paper, additional baselines of open-sourced large models for robot learning such Octo [1] can be evaluated with the proposed idea. The reason behind this is that the diffusion policy is known to work better for single tasks than multi-tasking. To enable diffusion policy to be a multi-task learner, Octo with diffusion policy as its action head is one of the popular models for such purpose;
 - In the continual learning experiment, does the size of the newly added expert and router ( number of active parameters) influence the performance?
 - Figure 5 is hard to read. The minor improvement of having mutual information loss renders it less necessary. Moreover, the three tasks here, e.g., "Lift" and "Square," share similar skills with "Can," such as picking and lifting, etc. That means the claimed effect of mutual loss to enforce specific tasks to specific experts is less sensible.
 - The last question is for the task transfer. As task transfer fine-tuning is similar to continual learning except for the frozen experts, it would be less confusing for readers if such design differences could be better motivated and explained.

[1] Team, Octo Model, et al. "Octo: An open-source generalist robot policy." arXiv preprint arXiv:2405.12213 (2024).

[2] Liu, Shiqi, et al. "Continual vision-based reinforcement learning with group symmetries." Conference on Robot Learning. PMLR, 2023.

**Quality Of The Limitations Section:**

2

**Questions For Rebuttal:**

See the list of weaknesses above.

**Robotics Focus:**

4

**Summary Of Paper:**

The authors propose to adopt a Mixture-of-Expert layer in the diffusion policy for high computation efficiency and continual learning. The introduced model is called sparse diffusion policy, featured by its sparsity and reusaibility for multi-task learning, continual learning and efficient task transfer. Experiments in both 2D/3D simulation and real robot verify the claimed properties.

**Summary Of Recommendation:**

Though this is an interesting idea, I vote for weak reject due to the less convincing experimental results.

---

### Decision · Program_Chairs · 2024-09-04

**Decision:**

Accept

**Comment:**

The authors have provided thorough responses with additional experiments  to the questions and concerns of the reviewers. The final scores are 1 weak accept, 1 accept and 1 strong accept
------
The paper "Sparse Diffusion Policy: A Sparse, Reusable, and Flexible Policy for Robot Learning" presents an innovative approach to multitask and continual learning in robotics using a mixture of experts within a diffusion policy. The strengths of this work lie in its novel idea and comprehensive experimentation demonstrating its feasibility and potential. However, the paper has several weaknesses, including concerns about memory consumption, the limited variety of tasks in experiments, and insufficient clarity in certain methodological details, making the experimental results less convincing.